# Patients with Pulmonary Metastases from Head and Neck Cancer Benefit from Pulmonary Metastasectomy, A Systematic Review

**DOI:** 10.3390/medicina58081000

**Published:** 2022-07-27

**Authors:** Georg Schlachtenberger, Fabian Doerr, Hruy Menghesha, Patrick Lauinger, Philipp Wolber, Anton Sabashnikov, Aron-Frederik Popov, Sascha Macherey-Meyer, Gerardus Bennink, Jens P. Klussmann, Thorsten Wahlers, Khosro Hekmat, Mathias B. Heldwein

**Affiliations:** 1Department of Cardiothoracic Surgery, Heart Center, University Hospital Cologne, Kerpener Strasse 62, 50937 Cologne, Germany; fabian.doerr@uk-koeln.de (F.D.); hruy.menghesha@uk-koeln.de (H.M.); anton.sabashnikov@uk-koeln.de (A.S.); gerardus.bennink@uk-koeln.de (G.B.); thorsten.wahlers@uk-koeln.de (T.W.); khosro.hekmat@uk-koeln.de (K.H.); mathias.heldwein@uk-koeln.de (M.B.H.); 2School of Medicine, University of Cologne, Albertus-Magnus-Platz, 50923 Cologne, Germany; lauinger.patrick@gmail.com; 3Department of Otorhinolaryngology, Head and Neck Surgery, Medical Faculty, University of Cologne, Kerpener Strasse 62, 50937 Cologne, Germany; philipp.wolber@uk-koeln.de (P.W.); jens.klussmann@uk-koeln.de (J.P.K.); 4Department of Cardiothoracic Surgery Helios Klinikum Siegburg, Ringstraße 49, 53721 Siegburg, Germany; aron-frederik.popov@helios-gesundheit.de; 5Department of Cardiology, Heart Center, University Hospital Cologne, Kerpener Strasse 62, 50937 Cologne, Germany; sascha.macherey-meyer@uk-koeln.de

**Keywords:** pulmonary metastasectomy, lung metastases, immunotherapy, head and neck cancer, metastatic head and neck cancer

## Abstract

*Background and Objectives*: The incidence of distant metastases in patients with head and neck cancer (HNC) is approximately 10%. Pulmonary metastases are the most frequent distant location, with an incidence of 70–85%. The standard treatment options are chemo-, immuno- and radiotherapy. Despite a benefit for long-term survival for patients with isolated pulmonary metastases, pulmonary metastasectomy (PM) is not the treatment of choice. Furthermore, many otorhinolaryngologists are not sufficiently familiar with the concept of PM. This work reviews the recent studies of pulmonary metastatic HNC and the results after pulmonary metastasectomy. *Materials and Methods*: PubMed, Medline, Embase, and the Cochrane library were checked for the case series’ of patients undergoing metastasectomy with pulmonary metastases published since 1 January 2000. *Results:* We included the data of 15 studies of patients undergoing PM. The 5-year survival rates varied from 21% to 59%, with median survival from 10 to 77 months after PM. We could not identify one specific prognostic factor for long-term survival after surgery. However, at least most studies stated that PM should be planned if a complete (R0) resection is possible. *Conclusions:* PM showed reliable results and is supposedly the treatment of choice for patients with isolated pulmonary metastases. Patients not suitable for surgery may benefit from other non-surgical therapy. Every HNC patient with pulmonary metastases should be discussed in the multidisciplinary tumor board to optimize the therapy and the outcome.

## 1. Introduction

Head and neck cancer (HNC) is a generic term for carcinomas of the oral cavity, the tongue, the larynx, and the pharynx. HNCs are the ninth most common malignancy, with high mortality rates [1,2]. More than 90% of HNC are squamous cell carcinomas (SCC), which originate from the epithelium of the mucosal lining of the upper aerodigestive tract [2]. The gold standard treatment for the management of these tumors is surgery, although radiotherapy, and other treatments play a role in the management of these conditions [3,4,5].

Interestingly, distant metastases in HNC occur at a lower rate than other malignancies, such as breast cancer, melanoma, or lung cancer [6]. The incidence of distant metastases in HNC is approximately 10% [6,7,8,9]. Patients with metastatic HNC are generally considered to have a poor prognosis [8]. Metastases of HNC seem to occur most frequently in the lungs with approximately 70–85%, followed by bones (15–39%) and liver (10–30%), respectively [10,11]. This frequent occurrence in the lungs is most likely due to the unique characteristics of the pulmonary system [1,12,13,14,15,16,17,18,19]. Specifically, the lungs receive the entire cardiac output every minute in the low-pressure system. As a result, the lungs have the densest capillary bed in the whole circulation and are the first reservoir of most lymphatic drainage. There are several ways to treat pulmonary metastases in HNC patients. Patients may receive chemo- and immunotherapy, checkpoint inhibitors, and radiation therapy or stereotactic body radiation (SBRT) [20,21]. In contrast, pulmonary metastasectomy (PM) has been performed in selected patients with isolated pulmonary metastases but is momentarily not the standard of care for pulmonary metastases. Recent studies showed the following mean survival rates of patients, according to their treatment:(1)Chemo- and immunotherapy and checkpoint inhibitors—7 to 14 months [21,22,23,24,25,26];(2)Radiotherapy and SBRT—approximately 21 months [27,28,29,30];(3)Pulmonary metastasectomy—10 to 77 months [31,32,33,34,35,36,37,38,39,40,41,42,43,44,45].

According to those studies, patients with pulmonary metastatic HNC undergoing PM tended to have better overall survival rates than patients receiving non-surgical therapy.

At least if PM is performed under specific criteria [46]:(1)The primary tumor needs to be treated curatively;(2)Distant metastases have to be ruled out with confidence;(3)R0 resection should be possible with adequate pulmonary reserve.

### Objective

While PM is mentioned in the guidelines on HNC, no direct recommendations are given to otorhinolaryngologists for the specific treatment of pulmonary metastatic HNC [5,47,48]. However, many otorhinolaryngologists are not sufficiently familiar with the concept of PM. Therefore, we reviewed recent studies on pulmonary metastatic HNC and their results after pulmonary metastasectomy in order to identify independent factors, which could be used to stratify patients who could significantly benefit from PM compared to non-surgical therapy.

## 2. Methods

### Study Inclusion and Exclusion Criteria

This systematic review followed the preferred reporting items for systematic reviews and meta-analyses (PRISMA) guidelines [49]. PubMed, Medline, Embase, and the Cochrane library were queried for the case series’ of patients undergoing metastasectomy with pulmonary metastases published since 1 January 2000.

Since the therapy in older studies sometimes deviates considerably from the current guidelines. Three independent investigators assessed study quality and bias risk (GS, PL, and SM-M).

The combination of the terms: (metastasectomy), (lung resection), (metastases), (pulmonary lesions), (head cancer), (head carcinoma), and (neck cancer and neck carcinoma) yielded 1036 results. Reviews and technical and pharmaceutical publications were excluded. In addition, we included publications in English and German with at least 20 surgically treated patients and metastasectomy as a curative approach.

We extracted data on patient population, resection type, status, method, postoperative complications, overall survival, recurrence rates, neo-adjuvant or adjuvant therapies, and long-term survival prognosis factors. The article selection process is depicted in Figure 1.

## 3. Results

We identified 15 studies on pulmonary metastasectomy in HNC that were carried out between January 2000 and February 2021 (see Table 1) [31,32,33,34,35,36,37,38,39,40,41,42,43,44,45].

### 3.1. Perioperative Data

The surgical approach, perioperative mortality, number of patients, and year of publication are presented in Table 1. The proportion of patients with a singular metastasis was between 15% and 87% [31,32,33,34,35,36,37,38,39,40,41,42,43,44,45]. Complete resection (R0) was achieved in 54–100% of patients in these studies.

In approximately 60% of operations, sublobular resection was preferred to anatomical resection (segmentectomy, lobectomy, bilobectomy, pneumonectomy). In addition, video-assisted thoracoscopic surgery (VATS) was performed in 57% of operations [31,32,33,34,35,36,37,38,39,40,41,42,43,44,45]. PM was associated with low perioperative mortality (0–3.7%) [31,32,33,34,35,36,37,38,39,40,41,42,43,44,45]. PM mostly led to mild complications with a morbidity rate between 0 and 14.4% [31,32,33,34,35,36,37,38,39,40,41,42,43,44,45].

### 3.2. Long-Term Survival after Pulmonary Metastasectomy

The long-term survival rates after PM are presented in Table 2.

The present studies showed median survival rates between 10–77 months [31,32,33,34,35,36,37,38,39,40,41,42,43,44,45]. The 3- and 5-year survival rates after PM were 7–67% and 21–59%, respectively [31,32,33,34,35,36,37,38,39,40,41,42,43,44,45]. The poorest long-term survival rates were reported by Mochizuki et al., with 3-year survival rates of 7% and a median survival rate of 10 months [31].

All other studies showed more favorable survival rates. In three cohorts, a proportion of ≥50% of patients showed survival exceeding 60 months [33,34,39]. The three most populous studies had 5-year survival rates between 21% to 36%. Disease-free interval (DFI) was reported in most studies. DFI is defined as the time between the surgery of the primary tumor and the occurrence of the pulmonary metastases. It is defined as the time between primary tumor and the occurrence of the first pulmonary metastasis. The DFI across studies was 12 to 45 months [31,32,33,34,35,36,37,38,39,40,41,42,43,44,45]. Patients with pulmonary metastatic HNC may also develop recurrent pulmonary metastases months or even years after initial PM [31,32,33,34,35,36,37,38,39,40,41,42,43,44,45]. It has already been shown for different primary tumor entities that redo pulmonary metastasectomy provides an overall survival benefit [50,51,52,53].

### 3.3. The Importance of Tumor Histology

Most head and neck cancers are squamous cell carcinomas (HNSCC) [12]. Younes et al. showed the prognostic relevance of primary tumor histology. They demonstrated that patients with HNSCC showed significantly poorer long-term survival rates than those with adenocarcinoma (AC) [54]. Mochizuki et al. showed devastating results for patients with HNSCC after PM, with a three-year survival rate of 7% [31]. Haro et al. showed better results for any other tumor histology than HNSCC, with three-year survival rates of 53% [33]. Winter et al. compared patients’ median survival after PM with adenoid cystic carcinoma (ACC) and HNSCC as primary tumor histology. They demonstrated a significantly longer median survival for ACC patients (43.5 months) compared to patients with HNSCC (15.2 months) (*p* > 0.0001) [35]. Smoking is a predominant risk factor for HNC, as well as for lung cancer. Not surprisingly, patients with HNC have a three to six times higher probability of also developing concomitant lung cancer than the general population due to the overlap of risk factors [55]. Therefore, it is essential to differentiate pulmonary metastases from primary lung cancer from a treatment perspective. Approximately 5% of pulmonary metastases turn out to be primary lung cancer after pathological examination [35,56,57]. The histology of HNSCC and lung cancer are so similar that differentiation is not usually promising by regular histopathological examinations [57,58]. Thus, differentiation is only possible by molecular pathological examination [59]. These pathological examinations are complex and can only be performed in high-volume centers. One possibility is that the human papillomavirus (HPV) evidence might be proof of pulmonary metastases from HNC, since expressing the expression of HPV in lung cancer is rather unlikely [60,61]. Moreover, p53 measurements might also help to differentiate metastases from secondary solitary tumors.

### 3.4. Prognostic Factors for Long-Term Survival after Metastasectomy

Independent risk factors are presented in Table 3. PM should only be considered after the multidisciplinary tumor board discussion (MDT). Hence, identifying favorable prognostic factors is relevant for evaluating long-term survival and selecting patients who would benefit from PM or non-surgical therapy, respectively. The International Registry of Lung Metastases (IRLM) included 5206 patients with pulmonary metastases from various primary epithelial and non-epithelial tumor entities [50]. They demonstrated that:(1)≤3 number of metastases;(2)R0 resection status;(3)Longer DFI;

were independent factors for a better long-term survival after PM [50]. However, as groundbreaking as the IRLM study was, the authors did not differentiate between different primary tumors developing pulmonary metastases. Consequently, those results are excluded from our study. Five of the included studies did not perform multinomial regression analysis [31,32,33,34,39], whereas nine studies performed multinomial regression analysis to identify independent prognostic factors [31,32,33,34,35,36,37,38,39,40,41,42,43,44,45] and were evaluated for prognostic factors here:

Affection and further therapy: Unilateral or bilateral pulmonary metastases and neo- and neo-adjuvant therapy were irrelevant prognostic factors for long-term survival after metastasectomy, according to the studies [31,32,33,34,35,36,37,38,39,40,41,42,43,44,45]. The overall number of metastases was classified as prognostically less relevant in four publications [35,36,37,40]. In contradiction, Younes et al. demonstrated statistical significance for the number of metastases [54]. According to Winter et al., patients with single metastasis survived longer than patients with multiple metastases, although without statistical significance (20.4 vs. 16 months) [35]. Oki et al. and Dudek et al. indicated that the lesion size of the pulmonary metastases was an independent risk factor of long-term survival [43,44]. Oki et al. and Yotsukura and colleagues showed that DFI was a significant prognostic factor for long-term survival [42,43].

Gender: The influence of gender as a prognostic factor is discussed and disputed in the literature. Shiono et al. showed that the male gender was an independent predictor of poor long-term survival after PM [36,37]. However, most included studies demonstrated that gender was not a prognostic factor for long-term survival [35,40,54].

Location of the primary tumor: Shiono et al. showed that the localization of the primary HNC was an independent risk factor for long-term survival [36]. They demonstrated that patients with metastatic HNC from the oral cavity had the poorest long-term survival rates [36]. Dudek et al. exhibited that primary tumor localization had no independent influence on the long-term prognosis [44].

Lymph node involvement: Shiono et al. demonstrated that intrathoracic lymph node affection was an independent risk factor for a shorter long-term survival [36]. No other study investigated intrathoracic lymph node affections independently.

Age: Yotsukura et al. demonstrated that old age was an independent risk factor for shorter long-term survival. According to the multivariate analysis, they demonstrated that patients > 70 years had a poorer prognosis [42].

Resection status: Winter et al. and Shiono et al. showed that R0 resection was a significant factor for long-term survival [35,37]. Nevertheless, Younes et al. reported no significant effect of a complete metastasectomy, but patients with R0 resection survived longer than with R1 or R2 resection (42.8 vs. 18.6 months) [54].

The independent prognostic factors of the studies were incohesive. Depending on the study, age, gender, tumor histology, HNC localization, DFI, the number of metastases, size of the pulmonary lesions, and resection status may have a prognostic influence on the long-term survival [31,32,33,34,35,36,37,38,39,40,41,42,43,44,45]. Summing up, no apparent prognostic factors could derive from these heterogeneous patients.

## 4. Discussion

Pulmonary metastasectomy is a reliable treatment for selected patients with isolated pulmonary metastatic head and neck cancer. However, not all patients are entirely suitable for surgery. For example, patients with other distant metastases, >3–5 pulmonary metastases, or inadequate pulmonary reserves may not be entirely suitable for PM. Multimodal therapy of metastatic HNC, therefore, includes chemo- and immunotherapy and checkpoint inhibitors, as well as radiation therapy, in addition to PM.

### 4.1. Medical Treatment

For decades, the only medical treatment option for patients with metastatic HNC was a platinum-based therapy [62,63]. Since immunotherapy and monoclonal antibodies were introduced for metastatic HNC patients in the mid-2000s, overall survival rates have been substantially prolonged [21,22,24,64]. Thus, it is clear that immunotherapy has also found its way into the guidelines for treating HNC [5]. Currently, there are different treatment options for metastatic HNC depending on the patients’ targets and physical condition [5]. One of the first-line therapy options is the EXTREME regimen consisting of chemotherapy (platinum-based and fluorouracil) combined with the monoclonal antibody cetuximab [24,25,65]. The EXTREME regimen is replaced by the more effective and better-tolerated TPex regimen as first-line therapy (docetaxel–platinum–cetuximab) [26,66,67]. Moreover, there is the fully human IgG4 PD-1 inhibitor nivolumab, which restores antitumor immunity. According to the latest guidelines, nivolumab is mainly used for local recurrence and less for treating distant metastases [21,64]. Nivolumab is currently the second-line therapy after pembrolizumab + platinum-based, the EXTREME regimen, and TPex regimen.

The median survival of patients treated with nivolumab, pembrolizumab, afatinib, the EXTREME regimen, or the TPex regimen was between 7 and 14.5 months [21,22,23,25,26]. In addition, the applied immunotherapies showed significantly better survival rates than the platin-based treatment [21,22,23,64], with a median survival rate of 5.1 to 7.4 months.

The high discrepancy in survival rates between patients after PM and patients receiving medical treatment is not the least since patients subjected to immunotherapy could no longer undergo PM for several reasons. These include the number or the size of the metastases, extrapulmonary metastases, and if patients are not suitable for PM due to their inadequate pulmonary reserve.

### 4.2. Radiation Therapy

Radiation therapy is now available for decades as a supplemental or alternative therapy stereotactic body radiation (SBRT) and is frequently regarded as an option in cases with arguments against surgery including compromised physical condition, unfavorable central location pulmonary nodule, or previous PM [27,28,29,30]. Ricco et al. treated patients with pulmonary metastases from different primary tumors with SBRT. A total of 51 of 577 were patients with metastatic HNC. They demonstrated a median survival of 37 months for HNC patients [27]. Chai et al. included 44 HNC patients in their analysis. After SBRT, the authors showed a median survival of 26 months [28]. Finally, Pasalic et al. and Bates et al. demonstrated one-year survival rates of 75% and 78% and two-year survival rates of 62% and 43%, respectively [29,30]. Ricco et al. and Chai et al. demonstrated that smaller pulmonary lesion size was associated with prolonged survival [27,28].

### 4.3. Is Pulmonary Metastasectomy an Option?

The critical factor against surgery is the lack of randomized controlled trials (RCT) on PM. Apart from the absence of RCT, few retrospective studies compare PM with non-surgical therapy. The only RCT by Treasure et al. was stopped due to the insufficient number of recruited patients [68]. Further compounding this imbalance, PM is a marginalized treatment option in most studies dealing with metastatic HNC [6]. Sekikawa et al. included 402 patients with HNC in their retrospective analysis [9]. A total of 37 (9.2%) of those patients developed distant metastases, 33 (8.2%) showed pulmonary metastases, and 12 (3%) would have been suitable for PM, but only 5 patients underwent PM [6]. Kang et al. included 779 patients in their analysis, 98 patients (12.6%) had distant metastases, 50 (6.4%) pulmonary metastases, and 26 (3.4%) isolated pulmonary metastases, but only 13 (1.7%) underwent PM [9]. Kang et al. included 779 patients in their analysis, 98 patients (12.6%) had distant metastases, 50 (6.4%) pulmonary metastases, and 26 (3.4%) isolated pulmonary metastases, but only 13 (1.7%) underwent PM [69]. However, the studies included the overall survival rates after PM were 10 to 77 months [31,32,33,34,35,36,37,38,39,40,41,42,43,44,45]. Therefore, it should be considered a reliable treatment option.

### 4.4. The Surgical Treatment of Pulmonary Metastases

Although survival improved due to modern immunotherapy, it should be considered that patients with isolated pulmonary metastases may benefit substantially from pulmonary metastasectomy. Stereotactic body radiation should also be considered an option in cases with arguments against PM [27]. A significant advantage of PM over SBRT is that surgery allows histological material to be obtained.

PM should only be considered if a complete (R0) resection is within the realm of possibility [35,37], if less than three pulmonary metastases are diagnosed [38,54] if the lesion size is smaller than 1.4 cm [43,44], and if there are only unilateral metastases [35]. Several studies indicated that a longer DFI resulted in a longer overall survival after PM [42,43,54]. There are several reasons why patients with a longer DFI benefit more from PM. It can be speculated that HNC patients with a longer DFI tend to have a less aggressive primary tumor [38,45]. These differences in DFI are not the least due to the different histology. Patients with SCC tend to have shorter DFI than patients with histological evidence of adenocarcinoma or adenoid cystic carcinoma. Furthermore, the timing of surgery for the primary HNC significantly impacts the DFI. Nevertheless, patients with a shorter DFI should also be subjected to PM. Nevertheless, patients with a shorter DFI should also be subjected to PM [34,70]. The most commonly used procedure for pulmonary metastasectomy is sublobar resection. It is well accepted as an appropriate procedure because of its reduced invasiveness and ability to preserve lung tissue while maintaining a sufficient safety distance. Anatomical resections, such as segmentectomy or even lobectomy, provide radicality and might be an option based on the size and localization of the pulmonary lesions but at the cost of healthy lung parenchyma. However, pneumonectomy should be excluded from the therapy, due to its 5–10% rate of serious perioperative complications and its high rate of postoperative morbidity [71]. Surgeons have the option of a thoracotomy or minimally invasive approaches via VATS and, more recently, robotic-assisted thoracic surgery (RATS). The advantage of lateral thoracotomy is a clear view of the operating field. Furthermore, entire manual palpation of the lung is feasible. Hence, a digital detection of metastases that remained invisible in previous CT scans might be possible [39,72,73,74]. Conversely, VATS has lower complication rates and a lower operative trauma. Furthermore, VATS may provide superior cosmetic results and earlier discharge from the hospital. In recent years, RATS has also become increasingly established in thoracic surgery [75]. RATS provides a three-dimensional view for the surgeon and only requires exceedingly tiny incisions for the surgical approach. Therefore, the advantages of lateral thoracotomy can almost be combined with the advantages of VATS. However, many hospitals are not yet equipped with robotic devices, so this approach is not yet feasible for every department [76,77,78]. As with all malignancies, HNC patients may also develop recurrent pulmonary metastases months or years after PM. For this reason, active surveillance, in addition to the tumor follow-up of the primary HNC, should be implemented within the first two years after PM. The appropriate screening method for pulmonary metastases are CT-scans of the thorax at six-month intervals within the first 24 months [13]. In addition to imaging techniques, an analysis of tumor markers could provide information about recurrences. Hundsdorfer et al. analyzed tumor markers in the context of an SCC of the oral cavity [79]. Patients with elevated “uPA” (urokinase-type plasminogen activator, threshold: 4.58 ng/mg protein) and elevated “PAI-1” (plasminogen activator inhibitor type 1, threshold: 106.3 ng/mg protein) levels were more likely to develop recurrences [79]. If patients develop recurrent ipsilateral pulmonary metastases, redo PM could provide overall survival benefits [50,51,52,53], naturally associated with the increased mortality and morbidity risks of a redo operation. Nevertheless, this procedure usually is quite tolerable for patients.

### 4.5. Which Therapy for Which Patient?

Despite significant advances in treating metastatic head and neck cancer in recent years, the overall survival remains poor. The median survival of patients undergoing pulmonary metastasectomy was 10–77 months [31,32,33,34,35,36,37,39,40,41,42,43,44,45,54]. The median survival of patients after immunotherapy was 7–10 months [21,22,23,25]. There is a wide variation in survival rates between studies. As already described, this is due to various reasons, such as the histology of the primary tumors. In addition, another confounding factor is the surgery performed. In the work of Mochizuki et al., for example, patients who underwent pneumonectomy were also included [31]. This is an operation that should not be performed because the mortality and morbidity are out of proportion to the benefit for the patient [71].

However, patients receiving non-surgical therapy are vastly different. Those patients usually are unsuited to undergo PM due to many distant metastases. Ferris et al. treated patients with tumor progression or recurrence within six months with nivolumab, regardless of the number or the size of the metastases [21]. Cohen et al. and Vermorken et al. used the same criteria in treating patients with pembrolizumab and Cetuximab, respectively [23,25]. Guigay et al. treated all patients with the TPex regimen regardless of the localization of the distant metastases [26].

In order to decide which patient might benefit from which therapy, all HNC patients with pulmonary metastases should always be discussed in the MDT comprising of thoracic surgeons, otorhinolaryngologists, medical oncologists, radiologists, pathologists, and other clinicians.

## 5. Conclusions

Patients with pulmonary metastatic head and neck cancer achieved an average survival rate between 10 to 77 months after pulmonary metastasectomy. However, only patients with isolated pulmonary metastases benefit from pulmonary metastasectomy. Patients with distant metastases, multiple bilateral pulmonary metastases, or inadequate pulmonary reserve may benefit from chemo- or immunotherapy, checkpoint inhibitors, radiotherapy, or stereotactic body radiation but with shorter overall survival rates. More studies, particularly randomized control trials, are needed to stratify the patient population further. In conclusion, every patient with pulmonary metastatic head and neck cancer should be carefully staged and discussed in the MDT to optimize therapy and outcomes.

## Figures and Tables

**Figure 1 medicina-58-01000-f001:**
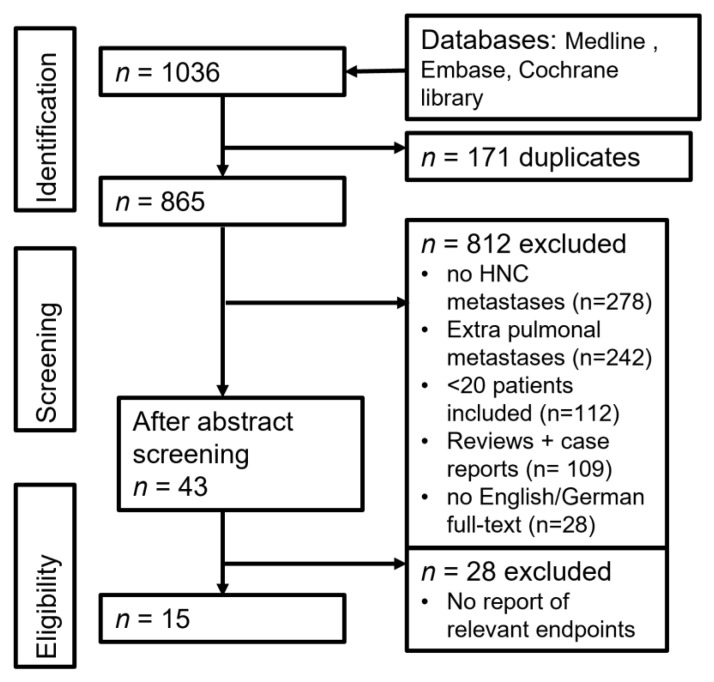
Flow-chart of literature research.

**Table 1 medicina-58-01000-t001:** Surgical approach.

Study	Patients	Singular Metastasis (%)	VATS (%)	Sublobular Resection (%)	R0 Resections (%)	Perioperative Mortality (%)
Mochizuki et al. [27]	23	87	/	9	69.6	/
Ma et al. [28]	28 (105)	/	/	65	/	0
Haro et al. [29]	21 (25)	80	/	68	/	/
Locati et al. [30]	20	15	0	69	54	/
Winter et al. [31]	67 (32)	37	15	78	80	3.7
Shiono et al. [32]	49	/	/	47	92	0
Shiono et al. [33]	114	74	/	33	90	/
Younes et al. [34]	104	37	0	52	70	0
Chen et al. [35]	20	45	0	80	100	0
Ichikawa et al. [36]	23	70	57	74	78.3	0
Miyazaki et al. [37]	24 (69)	79	36	56	88	0
Yotsukura et al. [38]	34	6	/	70	65	0
Oki et al. [39]	77	/	/	/	93	0
Dudek et al. [40]	44	48	18	73	/	2.3
AlShammari et al. [41]	56	/	22	/	/	0

Abbreviations: VATS: Video assisted thoracic surgery.

**Table 2 medicina-58-01000-t002:** Long-term survival rates after PM.

Study	Time of Resections	Median DFI (Months)	Median Survival (Months)	3-Years Survival (%)	5-Years Survival (%)
Mochizuki et al. [27]	1977–2003	12	10	7	/
Ma et al. [28]	1977–2008	/	/	61	32
Haro et al. [29]	1981–2008	17	/	53	50
Locati et al. [30]	1982–2001	45	/	/	53
Winter et al. [31]	1984–2006	19	21	/	21
Shiono et al. [32]	1984–2006	14	27	/	30
Shiono et al. [33]	1984–2006	16	26	/	27
Younes et al. [34]	1990–2008	/	33	/	36
Chen et al. [35]	1991–2007	27	/	/	59
Ichikawa et al. [36]	1991–2008	16	29	44	43
Miyazaki et al. [37]	1999–2009	25	/	67	/
Yotsukura et al. [38]	1986–2013	/	77	/	58
Oki et al. [39]	1992–2012	39	66	/	54
Dudek et al. [40]	2008–2018	/	29	/	41
AlShammari et al. [41]	2000–2016	/	/	79.5	71

Abbreviations: DFI: disease free interval.

**Table 3 medicina-58-01000-t003:** Multinomial regression analysis.

Prognostic Factors (Multivariate Analysis)	Winter et al. [31]	Shiono et al. [32]	Shiono et al. [33]	Younes et al. [34]	Ichikawa et al. [36]	Miyazaki et al. [37]	Yotsukura et al. [38]	Oki et al. [39]	Dudek et al. [40]
Age	n.s.	n.s.	n.s.	n.s.	n.s.	/	0.04	n.s.	n.s.
Gender	n.s.	/	0.004	n.s.	n.s.	/	n.s.	n.s.	n.s.
Localization of the primary tumor	/	/	<0.001	/	/	/	/	/	n.s.
Disease free interval	n.s.	0.04	n.s.	<0.05	n.s.	n.s.	0.02	0.04	/
Number of metastases	n.s.	n.s.	n.s.	<0.05	n.s.	n.s.	n.s.	n.s.	n.s.
Lesion size	n.s.	n.s.	n.s.	n.s.	n.s.	/	n.s.	0.01	0.0014
Affection (uni- vs. bilateral)	n.s.	n.s.	n.s.	/	/	/	/	/	n.s.
Anatomical vs. nonanatomic resection	/	n.s.	/	n.s.	n.s.	/	/	/	n.s.
Resection status	0.01	n.s.	0.009	n.s.	n.s.	/	/	/	/
Lymphnode affection (intrathoracic)	n.s.	/	0.009	/	/	/	/	/	/
Application of neo-/adjuvant therapy (metastasectomy)	/	/	/	/	n.s.	/	n.s.	n.s.	/

Abbreviations: n.s.: not significant.

## Data Availability

The data underlying this article will be shared on reasonable request to the corresponding author.

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
