# Peer review of "Patients with Pulmonary Metastases from Head and Neck Cancer Benefit from Pulmonary Metastasectomy, A Systematic Review"

_medicina, 2022, doi:10.3390/medicina58081000_

Round 1

Reviewer 1 Report

Dear Authors, I congratulate with you for this interesting and pleasant paper on a relevant topic. The text, apart for some punctuation errors, is well readable and offers a comprehensive perspective on PM for HNC metastases. Tables are informative. In view of the absence of more solid evidences in this context, I think this paper would be a useful tool to address patients to the best treatment option in a multidisciplinary setting.

However, from my point of view, there are some revision required before considering the paper acceptable on Medicina journal.

Major:

- I think the paper should be structured in a more adherent way to the PRISMA  protocol guidelines (https://prisma-statement.org/Extensions/Protocols). Several information are missing regarding the methodology of the study (selection process, selection criteria, number of independent reviewers).

- a figure depicting the article selection process should be inserted

Minor:

- pag. 2, line 49: thyroid cancer is not usually included in HNC definition.

- pag 2- line 96: "we only regarded the most recent published studies of patients receiving chemotherapy and radiation therapy in our analysis". But, isn't the review including only surgical series? I think this sentence is misleading.

- pag 7, line 161: "discussion" is repeated

- pag. 9, line 222: the "medical treatment" paragraph is too extensive considering that it does not represent the main topic of the paper. I suggest to make it more concise.

- pag 10, line 289: "Of course, patients have only a limited influence on the DFI." I personally do not understand this sentence and the relative justification for operating on patients with a short DFI. The whole concept should be reviewed or the concept expressed in a more solid way.

- I suggest reviewing punctuation and some text mistakes.

Author Response

Dear Authors, I congratulate with you for this interesting and pleasant paper on a relevant topic. The text, apart for some punctuation errors, is well readable and offers a comprehensive perspective on PM for HNC metastases. Tables are informative. In view of the absence of more solid evidence in this context, I think this paper would be a useful tool to address patients to the best treatment option in a multidisciplinary setting.

However, from my point of view, there are some revision required before considering the paper acceptable on Medicina journal.

Dear Reviewer, we cannot thank you enough for your recommendations. Overall, we have worked on all points. Therefore, the changes in the paper are marked in red.

Major:

Comment 1: I think the paper should be structured in a more adherent way to the PRISMA  protocol guidelines (https://prisma-statement.org/Extensions/Protocols). Several information are missing regarding the methodology of the study (selection process, selection criteria, number of independent reviewers).

Answer 1: This is a crucial recommendation. We worked revised the manuscript accordingly.

Changes 1: There are changes throughout the introduction. Furthermore, the whole method chapter was revised according to the Prisma protocol. Thank you again for pointing this out

Comment 2: a figure depicting the article selection process should be inserted

Answer 2: Thank you very much for this great suggestion. We developed a figure.

Changes 2: figure 1

Minor:

Comment 3: pag. 2, line 49: thyroid cancer is not usually included in HNC definition.

Answer 3: Thank you so much for pointing this out. The term was deleted.

Comment 4: pag 2- line 96: "we only regarded the most recent published studies of patients receiving chemotherapy and radiation therapy in our analysis". But, isn't the review including only surgical series? I think this sentence is misleading.

Answer 4: Thank you again very much for this recommendation. The sentence was deleted in the revised version.

Comment 5:  pag 7, line 161: "discussion" is repeated

Answer 5: Thank you for going through the article so carefully. Of course, the redundant word has been deleted.

Comment 6: pag. 9, line 222: the "medical treatment" paragraph is too extensive considering that it does not represent the main topic of the paper. I suggest making it more concise.

Answer 6: Thank you again for your marvelous recommendation. We shortened the paragraph. This should lead to an overall clarity.

Changes 6: The section on the medical treatment was shortened. (Page: 10; ll 230-252)

Comment 7: pag 10, line 289: "Of course, patients have only a limited influence on the DFI." I personally do not understand this sentence and the relative justification for operating on patients with a short DFI. The whole concept should be reviewed, or the concept expressed in a more solid way.

Answer 7: Thank you very much for this important recommendation.

Changes 7: We revised the entire section. (Page 12; ll: 291-298)

Comment 7: I suggest reviewing punctuation and some text mistakes.

Answer 7: Thank you very much for this recommendation.        

Changes 7: There are several changes throughout the manuscript.

Reviewer 2 Report

Although the review is interesting because it addresses an untreated problem elsewhere, it does not yet offer an answer for treatment options, but merely emphasizes the PM for patients eligible for metastasis surgery, criteria already present in the treatment guidelines of oncological diseases. The survival data of metastasis surgery on which the message of the review is based is burdened by some considerations:

About the selection criteria of patients eligible for pulmonary metastasectomy (PM) is important to specify what means "controlled or controllable" since it can alter the prognostic results (line 75)

Specify the duration of the Disease-free interval (DFI) (line 169)

Overall survival from 10 to 77 months depends on the surgical procedure adopted for PM, the overall survival data indicated rely on the procedure used. It is not possible to give a global survival datum, to provide indications to surgeons, if this results from multiple surgical procedures, each of which has different survival rates.

Consider the survival bias that exists between PM and the patient’s medical therapy in survival.

The overall survival depends not only on the PM, whose data is not unique, as it underlies several surgical procedures but also on the basic medical therapy of the patient.

The combination of the terms on PUBMED: (mastectomy), (lung resection), (metastasis), (lung lesions), (head cancer), (head cancer), (neck and neck cancer) gets 1204 results.

line 53 you should add: "the gold standard treatment for the management of these tumors is surgery, although radiotherapy, and other treatments play a role in the management of these conditions". and cite:doi: 10.3390/curroncol28040213. and doi: 10.3390/medicina57060563.

Thank You

Author Response

Although the review is interesting because it addresses an untreated problem elsewhere, it does not yet offer an answer for treatment options, but merely emphasizes the PM for patients eligible for metastasis surgery, criteria already present in the treatment guidelines of oncological diseases. The survival data of metastasis surgery on which the message of the review is based is burdened by some considerations:

Dear Reviewer, we cannot thank you enough for your recommendations. Overall, we have worked on all points. Therefore, the changes in the paper are marked in red.

Comment 1: About the selection criteria of patients eligible for pulmonary metastasectomy (PM) is important to specify what means "controlled or controllable" since it can alter the prognostic results (line 75)

Answer 1: Thank you for highlighting this ambiguity for the esteemed reader. We have rewritten the paragraph.

Changes 1 (page: 4; ll: 82): the primary tumor needs to be treated curatively

Comment 2: Specify the duration of the Disease-free interval (DFI) (line 169)

Answer 2: Thank you very much for reading our manuscript so carefully. Of course, a corresponding passage has been inserted.

Changes 2 (page: 6, ll: 137-139): Disease-free interval (DFI) was reported in most studies. DFI is defined as the time between the surgery of the primary tumor and the occurrence of pulmonary metastases.

Comment 3: Overall survival from 10 to 77 months depends on the surgical procedure adopted for PM, the overall survival data indicated rely on the procedure used. It is not possible to give a global survival datum, to provide indications to surgeons, if this results from multiple surgical procedures, each of which has different survival rates.

Answer 3: Thank you very much for pointing this out. In this work, we do not present our data but a review of other authors' data. Of course, you are correct that many confounding factors play a role in this large dispersion of survival data. For example, in some studies, pneumonectomy was performed for metastasectomy. An operation that would not be performed in our center under any circumstances, because the mortality and morbidity are out of proportion to the benefit for the patient.

To address your significant comment, we added a passage to the discussion.

Changes 3 (page 14; ll 337-341): As already described, this is due to various reasons, for example, the histology of the primary tumors included. In addition, another confounding factor is the surgery performed. In the work of Mochizuki et al., for example, patients who underwent pneumonectomy were also included. An operation that should not be performed, because the mortality and morbidity are out of proportion to the benefit for the patient

Comment 4: Consider the survival bias that exists between PM and the patient’s medical therapy in survival.

Answer 4: Thank you for this comment. You are addressing an elementary bias here. The patients scheduled for PM are handpicked. These are patients who meet the criteria that are addressed multiple times in the review. The patients who are assigned to drug therapy are virtually all patients who are not eligible for PM. This bias can only ever be "reduced" in retrospective studies. It can only be excluded if an RCT is performed, which is unlikely due to the points described. A section describing this bias was added to the chapter on medical therapy.

Changes 4 (pages 10; ll: 249-252): The high discrepancy in survival rates between patients after PM compared to patients receiving medical treatment is not the least because patients subjected to immunotherapy could no longer undergo PM due to several reasons. The number or the size of the metastases, extrapulmonary metastases, and if patients are not suitable for PM due to their inadequate pulmonary reserve

Comment 5: The overall survival depends not only on the PM, whose data is not unique, as it underlies several surgical procedures but also on the basic medical therapy of the patient.

Answer 5: Thank you for highlighting this proposal. We have put much effort into elaborating this recommendation in Response and Changes 3 and 4 of your revision.

Changes 5: no changes

Comment 6: The combination of the terms on PUBMED: (mastectomy), (lung resection), (metastasis), (lung lesions), (head cancer), (head cancer), (neck and neck cancer) gets 1204 results.

Answer 6: Thank you for your research. We performed the literature recherche on 10.02.2021. In the meantime, of course, other papers have appeared under these search criteria. Last but not least, our research group published an original paper on the treatment of HNC metastases in the meantime, which was picked up by an editorial.

Changes 6: no changes

Comment 7: line 53 you should add: "the gold standard treatment for the management of these tumors is surgery, although radiotherapy, and other treatments play a role in the management of these conditions". and cite:doi: 10.3390/curroncol28040213. and doi: 10.3390/medicina57060563.

Answer 7: Thank you very much for this suggestion, together with the literature. This section was, of course inserted into the manuscript.

Changes 7 (page 3; ll: 57-59): The gold standard treatment for the management of these tumors is surgery, although radiotherapy, and other treatments play a role in the management of these conditions

Round 2

Reviewer 1 Report

Dear Authors, I went through your revised version of the paper. From my point of view it can be accepted for publication.

Best regards.